# Cationic Surfactant-Modified *Tetraselmis* sp. for the Removal of Organic Dyes from Aqueous Solution

**DOI:** 10.3390/molecules28237839

**Published:** 2023-11-29

**Authors:** Huda M. Alghamdi, Khalid Z. Elwakeel

**Affiliations:** 1Department of Chemistry, Faculty of Mathematic and Natural Sciences, University of Lampung, Jl. Soemantri Brojonegoro No. 1, Bandar Lampung 35145, Indonesia; buhani@fmipa.unila.ac.id (B.); isti9515@gmail.com (I.); suharso@fmipa.unila.ac.id (S.); 2Department of Electrical Engineering, Faculty of Engineering, University of Lampung, Jl. Soemantri Brojonegoro No. 1, Bandar Lampung 35145, Indonesia; sumadi.1973@eng.unila.ac.id; 3Department of Medicine, Faculty of Medicine, University of Lampung, Bandar Lampung 35141, Indonesia; 4Department of Chemistry, College of Science, University of Jeddah, Jeddah 80327, Saudi Arabia; halghamdi3@uj.edu.sa; 5Environmental Chemistry Division, Environmental Science Department, Faculty of Science, Port Said University, Port Said 42522, Egypt

**Keywords:** *Tetraselmis* sp. algae, cethyltrimethylammonium bromide, adsorption, methylene blue, methyl orange

## Abstract

The modification of the Tetraselmis sp. algae material (Tetra-Alg) with surfactant Cethyltrimethylammonium Bromide (CTAB) yielded adsorbent Tetra-Alg-CTAB as an adsorbent of methyl orange (MO) and methylene blue (MB) solutions. The characterization of the adsorbent used an infrared (IR) spectrometer to identify functional groups and Scanning Electron Microscopy with Energy Dispersive X-ray (SEM-EDX FEI Inspect-S50, Midland, ON, Canada) to determine the surface morphology and elemental composition. Methyl orange and methylene blue adsorption on the adsorbent Tetra-Alg, *Tetraselmis* sp. algae-modified Na^+^ ions (Tetra-Alg-Na), and Tetra-Alg-CTAB were studied, including variations in pH, contact time, concentration, and reuse of adsorbents. The adsorption of MO and MB by Tetra-Alg-CTAB at pH 10, during a contact time of 90 min, and at a concentration of 250 mg L^−1^ resulted in MO and MB being absorbed in the amounts of 128.369 and 51.013 mg g^−1^, respectively. The adsorption kinetics and adsorption isotherms of MO and MB and Tetra-Alg, Tetra-Alg-Na, and Tetra-Alg-CTAB tend to follow pseudo-second-order kinetics models and Freundlich adsorption isotherms with each correlation coefficient value (R^2^) approaching 1. Due to the modification with the cationic surfactant CTAB, anionic dyes can be strongly sorbed in alkaline pH due to strong electrostatic attraction, while MB is more likely to involve cation exchange and hydrogen bonding. The reuse of Tetra-Alg-CTAB was carried out four times with adsorption percent > 70%, and the adsorbent was very effective in the adsorption of anionic dyes such as MO.

## 1. Introduction

Aquatic environmental problems, such as water that is not fit for consumption, have become very common nowadays. The problem is triggered by pollution originating from the accumulation of various kinds of waste. One sector that contributes to waste in the aquatic environment is the textile industry sector. Although the textile industry is one of the sectors that has a positive impact on supporting economic development, the textile industry also contributes a very detrimental negative impact to the environment due to the presence of waste products in the form of dye waste. This is supported by data; it is estimated that the textile industry consumes about 60% of the total dye produced, and about 10–15% of the remaining dye comes out as waste [1,2]. When the waste is discharged into the aquatic environment, the textile dyes contained in the waste will become more stable and more difficult to decompose due to the complex chemical structure formed [3].

Dye waste that enters water forms a more complex chemical structure, causing the dye to be difficult to decompose in water [4]. Types of synthetic dyes that are widely produced and used in the textile industry are thiazine dyes, such as methylene blue, and compounds with an azo group, such as methyl orange. Thiazine compounds have functional group characteristics, namely the presence of a group (C=N), and if there is an increase in the amount of thiazine dye in the water, it will reduce the level of dissolved oxygen in the water and damage aquatic life [5,6]. Methyl orange is one of the anionic dyestuffs with an azo group and its derivatives from the benzene group [7]. Azo compounds are compounds that have an N=N group, called the azo structure, and are most commonly found in waste. Because of their widespread usage, toxicity, carcinogenicity, and poor removal rates during wastewater treatment, most of these synthetic dyes pose a threat to the environment [8,9,10,11]. Additionally, when the amount of oxygen in the water drops, anaerobic organisms become more active, producing a product with an unpleasant odor [12].

Treating garbage that contains dyes before it is released into the environment is one of the best strategies to lower pollution and the spread of dyes in the environment. Synthetic dyes have been eliminated from industrial effluents using a variety of physical and chemical techniques, including adsorption, flocculation, coagulation, membrane filtration, photocatalytic degradation, and irradiation [13,14,15,16]. Of these several methods, the adsorption method is quite widely used. The adsorption process is easy to use, reasonably priced, and produces no environmentally hazardous byproducts [17,18]. The kind and compatibility of the adsorbent utilized have a major impact on the adsorption process’s outcome. A high adsorption rate and capacity are necessary for an efficient adsorbent in adsorption. It can also be utilized repeatedly, is environmentally friendly, and is chemically stable [19,20].

Currently, adsorption is being widely developed to reduce the contamination of synthetic dyes in liquid waste by using adsorbent base materials derived from organisms such as fungi, bacteria, yeast, and algal biomass [21,22,23]. Algal biomass is generally used as an adsorbent because it can be available in large quantities. In addition, it is an effective and relatively inexpensive adsorbent [20,24].

By nature, algal biomass is a highly potent adsorbent that can remove contaminants from organic substances, including dye-derived chemicals [19,22,25]. Nevertheless, several limitations, including their tiny size, low specific gravity, and susceptibility to degradation by other microbes, limit their capacity to bind these chemical substances [26,27]. Numerous attempts have been made to address these shortcomings, including improving the biomass of the algae using different modifying agents [20,21,28].

The capacity and quality of algal biomass can be increased by using adsorbent surface modifying agents such as cationic surfactant CTAB. Surface modification of algal biomass with cationic surfactants can make the adsorbent rich in positive charges so that it will be effective against adsorbates that tend to be negatively charged [29,30,31]. An adsorbent has an alkaline or alkaline earth cation exchange active group that can be swapped out for various cations, including cationic surfactants, to serve as a charge balancer [32,33,34].

A good sorbent, especially for large-scale applications, must be recyclable in order to be competitive. Both the demand for fresh sorbent and the issue of disposing of spent sorbent can be addressed via regeneration. Different regeneration techniques have been applied, with varying degrees of effectiveness. Solvent-washing, thermal, chemical, and electrochemical regeneration are some of these techniques [35]. Recently, upcycling of dye-loaded spent sorbent via an appropriate management pathway (anaerobic fermentation) for the development of a low-cost hydrogen production scenario was investigated [36]; the findings of that study inferred that the spent algal sorbent could be introduced as an effective substitute for biohydrogen production.

Based on this description, an adsorbent was created in this study using the cation exchange characteristics of CTAB surfactants and the active groups of algal biomasses to adsorb MO and MB dyes. The algal biomass of *Tetraselmis* sp. was modified for this purpose. Consequently, it is possible to produce an efficient adsorbent to adsorb colors in solution. Dye desorption and sorbent recycling are carried out after sorption kinetic and equilibrium investigations. The findings of this study may offer a way to stop dangerous dyes from spreading across the environment.

## 2. Results and Discussion

### 2.1. Adsorbent Characterization

Analysis of Tetra-Alg, Tetra-Alg-Na, and Tetra-Alg-CTAB (Figure 1) using an IR spectrometer showed that, on the three adsorbents, there was an absorption band at a wave number of 3695.61 cm^−1^ which indicated the presence of N-H stretching vibrations. In addition, at a wave number around 3425.58 cm^−1^, there is an absorption band indicating the presence of O-H stretching vibrations, and at 2931.80 cm^−1^ one from the C-H (alkyl) stretching vibration of aliphatic (-CH_2_) and the carbonyl group (C=O) at number waves around 1651.07 cm^−1^. The success of Tetra-Alg modification with CTAB is indicated by the appearance of an absorption band at wave number 1465.90 cm^−1^ originating from the methyl group contained in CTAB in Tetra-Alg-CTAB (Figure 1c), while in Tetra-Alg and Tetra-Alg-Na the absorption band does not appear.

Surface morphology and elemental constituents of Tetra-Alg, Tetra-Alg-Na, Tetra-Alg-CTAB were studied using SEM-EDX. The results of the analysis using SEM-EDX (Figure 2) showed that there were differences in morphology between Tetra-Alg, Tetra-Alg-Na, and Tetra-Alg-CTAB. In Figure 2a, Tetra-Alg morphology looks less contrasting and there are homogeneous grains, while Tetra-Alg-Na (Figure 2b) shows a less homogeneous morphology with more contrasting colors. The contrasting color difference comes from the presence of the metallic element of Na. Furthermore, in Tetra-Alg-CTAB) there are grains attached to the surface, which are homogeneous with less contrasting colors. This shows that the modification of Tetra-Alg with CTAB causes CTAB not to be absorbed in Tetra-Alg, but only to cover the algal cell wall [20,37].

The results of the analysis with surface morphology are supported by the results of the analysis with EDX. Based on EDX data on Tetra-Alg (Figure 2a), there are the elements C, N, and O derived from organic groups from algal biomass while in Tetra-Alg-Na, apart from the elements found in Tetra-Alg, other elements appear, that is, Na derived from NaCl. Furthermore, the EDX data on Algae-CTAB (Figure 2c) contained the elements of C, N, O, and Br. The presence of Br in Tetra-Alg-CTAB indicates that modification of CTAB surfactant on Tetra-Alg has been successfully carried out [37].

The surface area of the resin was calculated using the following Equation [38]:(1)$$A_{s}=\frac{10^{-20}}{M\;M_{W}}\;G\;N_{AV}\phi$$
where $A_{s}$ is the adsorbent surface area in m^2^ g^−1^, *G* the amount of methylene blue adsorbed (g) based on Langmuir adsorption capacity, *N_Av_* the Avogadro number (6.02 × 10^23^), *ϕ* the methylene blue molecular cross-section (197.2 Å), *M_W_* the molecular weight of methylene blue (373.9 g mol^−1^) and *M* the mass of adsorbent (g). The surface area of Tetra-Alg, Tetra-Alg-Na, and Tetra-Alg-CTAB was found to be 158.93, 184.64, and 161.97 m^2^ g^−1^, respectively.

### 2.2. pH_pzc_ of Adsorbent

Determination of the pH_pzc_ value was carried out to study the effect of pH on the surface charge of Tetra-Alg, Tetra-Alg-Na, Tetra-Alg-CTAB. In Figure 3, it can be observed that the pH_pzc_ value of the three adsorbents is at pH 8. The pH area below pH_pzc_ is positively charged, while the pH area above pH_pzc_ is negatively charged [19,39].

### 2.3. Effect of Adsorption Initial pH

Because pH changes the dissolved adsorbate speciation and the active surface charge of the adsorbent, it has a major effect on the adsorption process. In Figure 4, the relationship between initial pH (pH_0_) and % of adsorbed dye can be observed. The effect of interaction pH_0_ on % MO adsorbed (Figure 4) showed that % MO adsorbed at interaction pH 10 by Tetra-Alg-CTAB was the largest (99.32%) compared to Tetra-Alg (5.86%) and Tetra-Alg-Na (5.11%). This happens because of the modification with CTAB to obtain an adsorbent that is rich in positively charged active sites. Based on the pHpzc value (Figure 3), it can be stated that at pH 10 the adsorbent has a positive surface charge so that it can properly adsorb the anionic (negative) MO dye through electrostatic interactions, causing high adsorption [29,37].

In Figure 4, it can also be seen that the % MB adsorbed by Tetra-Alg and Tetra-Alg-Na in all pH ranges is almost the same. In Tetra-Alg-CTAB, the % MB adsorbed was relatively lower than Tetra-Alg and Tetra-Alg-Na. This happens because Tetra-Alg-CTAB is dominated by positive charge, so it is less effective in interacting with cationic MB. Therefore, the increase in MO removal after cationic surfactant modification can be explained by electrostatic interactions. Even at pH_0_ values = 3, the adsorption of MO remained significantly high due to the presence of the cationic surfactant, while MB adsorption is more likely to involve cation exchange and hydrogen bonding. The adsorption process mechanism may also be significantly influenced by other forces such as n–π stacking interactions and dipole-dipole hydrogen bonding [40]. It is also important to remember that some oppositely charged surface sites may have existed concurrently with the overall negatively or positively charged adsorbent surface charge. Also, in Figure 4, it can be observed that the highest adsorption occurred at pH 12 for Tetra-Alg, Tetra-Alg-Na, Tetra-Alg-CTAB with % MB adsorbed, respectively, being 97.41, 98.35, and 99.11%. However, at pH 12 there was high adsorption of MB by the three adsorbents; it is assumed that the interaction that occurs between the adsorbent and MB is a precipitation process caused by the high concentration of OH- in the solution under alkaline conditions [21].

### 2.4. Effect of Contact Time

The relationship between adsorption contact time on % MO and MB adsorbed by Tetra-Alg, Tetra-Alg-Na, Tetra-Alg-CTAB is shown in Figure 5. In Figure 5, it can be seen that the addition of relative adsorption interaction time did not increase the % MO and MB adsorbed. At the interaction time of 90 min, it was observed that the % MO adsorbed on Tetra-Alg, Tetra-Alg-Na, and Tetra-Alg-CTAB was 12.86, 14.97, and 99.17%, respectively. Then, at the same time, the contact time of 90 min, the % MB adsorbed by Tetra-Alg, Tetra-Alg-Na, and Tetra-Alg-CTAB was 93.38, 96.58, and 61.94%, respectively.

In Figure 5, it can be observed that, when the two data of MO and MB dyes are compared, MO dyes are more adsorbed on Tetra-Alg-CTAB while MB tends to be adsorbed by Tetra-Alg and Tetra-Alg-Na. This is in line with the data obtained on the effect of pH interactions (Figure 4). The interaction of MO with Tetra-Alg-CTAB occurs through electrostatic interactions because the adsorbent Tetra-Alg-CTAB is rich in positively charged active sites while MO is anionic [30,31].

### 2.5. Adsorption Kinetics

By utilizing the pseudo-second-order (PSO) kinetic model (Equation (2)) and the intra-particle diffusion (IPD) Model (Equation (3)), the adsorption kinetics parameters of MO and MB dyes were ascertained through an analysis of the data presented in Figure 5. The pseudo-second-order kinetic model assumes that the adsorption capacity is proportional to the number of active sites on the adsorbent. By applying the linear Equation (2),
(2)$$\frac{t}{q_{t}}=\frac{1}{k_{2}q_{e}^{2}}+\frac{t}{q_{e}}$$
where *q_t_* and *q_e_* (mg g^−1^) are total MO or MB adsorption capacity at time *t* and equilibrium, respectively, and *k*_2_ is the second-order rate constant, respectively, the results of the analysis of the adsorption kinetics data were obtained and are presented in Table 1.

From Table 1, it can be seen that each value of the linear correlation coefficient (*R*^2^) of the second-order model of the MO and MB dyes by Tetra-Alg, Tetra-Alg-Na, Tetra-Alg-CTAB is relatively the same. If the *k*_2_ values of both MO and MB dyes are compared, it can be seen that the MO adsorption rate is greater than MB. Furthermore, the adsorption rate of MO-dye on Tetra-Alg-CTAB is higher than that of Tetra-Alg and Tetra-Alg-Na.

Data on the effect of contact time (Figure 5) were also analyzed using the intra-particle diffusion (IPD) kinetic model developed by Morris and Weber [41,42]. The IPD kinetic model assumes that the diffusion process of an adsorbate, such as dye, to an adsorbent is controlled by a physical mechanism that has an important role in the adsorption process, as described by Equation (3).
(3)$$q_{t}=k_{id}t^{0.5}+C$$

The intra-particle diffusion rate constant is denoted by *k_id_* (mg g^−1^ min^−0.5^), whereas the constant *C* value (mg g^−1^) indicates the barrier to mass transfer in the boundary layer. The slope and intercept of the lines derived from plots of *q_t_* versus *t*^0.5^ were used to compute the values of *k_id_* and *C*. The results of the analysis of MO and MB adsorption data by Tetra-Alg, Tetra-Alg-Na, and Tetra-Alg-CTAB obtained from the completion of the IPD kinetic model are shown in Table 2 and Figure 6. 

It is evident from Figure 6 that every plot has two linear components. This pattern suggests the involvement of many adsorption models. The external mass transfer is shown by the pattern in the first linear portion, which spans an adsorption time of 0 to 60 min. The second linear section covers an adsorption period of 60–120 min, representing the pattern describing the intraparticle diffusion of the particles. The fact that the second linear portion does not pass through the origin (C ≠ 0) shows that intraparticle diffusion controls the rate of external mass transfer simultaneously with other steps [41,42].

Based on the observations in Figure 6, two steps describe the transfer of the dye from the solution to the external surface of the adsorbent and directed diffusion. Then, the step is continued from the dye to the active site of each adsorbent through the pore cavity and the active group. The adsorption mechanism can be explained in two different ways, namely the occurrence of diffusion through the pores of the adsorbent and electrostatic interactions in the presence of a positively charged functional group from the modification of Tetra-Alg to Tetra-Alg-CTAB [43]. This is in line with the results of the second-order pseudo-kinetic model analysis that has been discussed previously.

### 2.6. Adsorption Isotherm

The MO and MB adsorption isotherm models were studied by varying the dye concentration from 0–250 mg L^−1^, as shown in Figure 7. Based on the data in Figure 7, it can be observed that the adsorption of MO and MB by Tetra-Alg, Tetra-Alg-Na, and Tetra-Alg-CTAB increased with the increasing concentrations used. The data contained in Figure 7 were analyzed using the Freundlich (Equation (4)) and Langmuir (Equation (5)) isotherm models, as follows:(4)$$\log q_{e}=\log K_{F}+\frac{1}{n}\log C_{e}$$
(5)$$\frac{1}{q_{e}}=\frac{1}{q_{m}K_{L}C_{e}}+\frac{1}{q_{m}}$$
where *K_F_* is the Freundlich constant related to adsorption capacity (mg g^−1^) (L mg^−1^)^1/*n*^) and *n* is the adsorption intensity. *K*_*L*_ is the ratio of the adsorption and desorption rate in L mg^−1^, and *q*_*m*_ is the maximum adsorption capacity estimated by the Langmuir model in mg g^−1^.

The Freundlich isotherm model is an empirical formula used to describe adsorption at multilayer and heterogeneous systems [44]. The Langmuir adsorption isotherm model assumes that the adsorbent surface is uniform, there are a fixed number of active sites proportionate to the surface area, and the adsorption process is monolayer [45,46]. The results of the data analysis in Figure 7 are shown in Table 3.

Based on the data obtained in Table 3, it can be observed that the adsorption on MO and MB tends to follow the Freundlich adsorption isotherm model. This can be seen from the coefficient value (*R*^2^) on the Freundlich adsorption isotherm, which is closer to 1 when compared to the correlation coefficient value (*R*^2^) on the Langmuir adsorption isotherm. The Freundlich adsorption isotherm model assumes that the dye layer formed on the surface of the adsorbent is a multilayer [47]. From Table 3, it can be observed that the adsorption capacity of MO that was adsorbed by Tetra-Alg-Na and Tetra-Alg-CTAB was greater than that of MB. This happens because the interaction that occurs between MO with Tetra-Alg-Na and Tetra-Alg-CTAB tends to occur through electrostatic interactions, while MB is more likely to involve cation exchange [32,33].

Table 4 presents a comparison between the adsorption capacities of Tetra-Alg-CTAB for the MO and MB dyes and those of previously documented adsorbents. The information in Table 4 illustrates that Tetra-Alg-CTAB exhibits remarkable effectiveness as an adsorbent for the MO dye, especially when contrasted with other adsorbents featured in the same table. The comparison of sorption performances for Tetra-Alg-CTAB and with other algal biomasses clearly demonstrates the synergistic effect of precursors. The incorporation of the cationic surfactant CTAB in the algal biomass allows for the improvement of the sorption of MO. The cationic surfactant-modified biosorbent exhibited a notable increase in its ability to attract anionic dye molecules. This is due to the introduction of positive charges on the Tetra-Alg-CTAB surface, which enhances the adsorption of MO via electrostatic interaction, while MB sorption is more likely to occur through cation exchange.

### 2.7. Adsorbent Reuse

The reuse of Tetra-Alg, Tetra-Alg-Na, and Tetra-Alg-CTAB adsorbents to adsorb MO solution in several cycles is an important parameter in determining the quality of the adsorbent produced. The adsorbent can be reused several times to obtain the best quality from the adsorbent and economic efficiency.

In Figure 8 it can be seen that the reuse of Tetra-Alg-CTAB to adsorb MO showed effective results of more than 70% on four successive repetition cycles, of 99.97, 91.97, 81.41, and 71.90%. The reuse of Tetra-Alg and Tetra-Alg-Na adsorbents is not very effective. This is in line with the low % MO-dye adsorbed in the first cycle and continues to decrease until the fifth cycle. The adsorption yield decreased significantly in each iteration cycle. The decrease in adsorption ability occurs due to the decrease in the active site on the adsorbent caused by the desorption process carried out to release the adsorbed methyl orange dye [55].

## 3. Materials and Methods

### 3.1. Materials and Instruments

The algal biomass of *Tetraselmis* sp. was obtained from cultivation at the Center for Marine Cultivation in Lampung, Indonesia and prepared into algal biomass in the Inorganic chemistry laboratory of the Faculty of Mathematics and Natural Sciences, University of Lampung. MO and MB dyes (Appendix A) and other chemicals required for algal biomass modification and adsorption processes such as NaCl, NaNO_3_, CTAB, HCl, NaOH, citrate buffer, and phosphate buffer are analytical reagents (AR) grade and purchased from Pharmacopoeia European.

The characterization of the adsorbent was carried out with an IR spectrometer (Shimadzu IRPrestige-21, Shimadzu Corporation, Tokyo, Japan) to identify the functional groups of the adsorbent. Surface morphology analysis of the adsorbent was carried out using Scanning Electron Microscopy with Energy Dispersive X-ray (SEM-EDX FEI Inspect-S50, Midland, ON, Canada).

### 3.2. Adsorbent Preparation

The Tetra-Alg biomass was air-dried for 3 days and continued in the oven at 40 °C for 3 h, then ground using a grinder to form a fine powder with a size of 100 mesh. A total of 5 g of Tetra-Alg was added to 100 mL of 0.1 M NaCl in a 250 mL Erlenmeyer flask. The mixture was stirred using a shaker for 1 h, then allowed to stand for 24 h with the Erlenmeyer mouth closed. After being separated, the resulting precipitate was washed with distilled water until neutral. After that, the precipitate dried at room temperature until the weight was constant (Tetra-Alg-Na). Then, 2 g of Tetra-Alg-Na was mixed with 200 mL 14 mmol L^−1^ CTAB solution in an Erlenmeyer flask and put into a water bath at 50 °C under stirring. Then, the resulting precipitate was washed with distilled water until neutral and dried at room temperature to a constant weight to produce Tetra-Alg-CTAB.

### 3.3. Determination of pH_pzc_ of Adsorbent

A total of 0.05 g of adsorbent was mixed with 10 mL of 0.1 M NaNO_3_ solution with a pH varying between 3 to 12. pH adjustment can be carried out by adding 0.1 M NaOH for alkaline conditions and 0.1 M HCl for acidic conditions. The solution was then stirred using a stirrer for 24 h. The supernatant was then poured, and the pH was measured. The pH_pzc_ value was obtained from the plot of the pH of the initial solution against the pH of the supernatant [53].

### 3.4. Dye Adsorption Batch Experiments

MO and MB adsorption studies were carried out using the batch method. The adsorbates used were MO and MB dyes. Adsorbate mother liquor (1 g L^−1^) is prepared by dissolving a certain amount of adsorbate in distilled water. Then, the mother liquor is diluted to the desired concentration. The parameters studied include the determination of adsorption kinetics and isotherms. The standard curve between absorbance (A) and dye concentration (*C*_0_, mgL^−1^) was measured with a UV-Vis spectrophotometer (Agilent Cary 100, Markham, ON, Canada) at the maximum absorption wavelength of the dye (MO max = 465 nm; and MB max = 664 nm). MO or MB dyes were mixed with the adsorbent at a pH varying between 3–12, during a contact time between 0–90 min, and at a dye concentration varying from 0–250 mg L^−1^. Adsorption was carried out in a shaker incubator at a temperature of 25 °C at a speed of 200 rpm. The mixture between the filtrate and the precipitate was separated by centrifugation, and the filtrate was analyzed using a UV-Vis spectrophotometer. The level of MO or MB adsorbed by the adsorbent is determined by calculating the adsorbate adsorbed per unit mass of adsorbent, and the percentage of dye absorbed by the adsorbent is calculated using the following equations:(6)$$q_{e}=\frac{\left(C_{0}-C_{e}\right)}{m}\times V$$
(7)$$q_{t}=\frac{\left(C_{o}-C_{t}\right)}{m}\times V$$
(8)$$Adsorption\;(\%)=\frac{\left(C_{o}-C_{t}\right)}{C_{o}}\times 100$$
where *C_o_* is the initial dye concentration, *C_e_* is the equilibrium concentration, *C_t_* is the concentration at a certain time *t* of dye solution (mg L^−1^), *m* is the mass of adsorbent (g), *V* is the volume of the solution (L), *q* is the amount of dye adsorbed per unit mass (mmol g^−1^), and *Adsorption* (%) is the percentage of the dye removal. All adsorption tests were performed in triplicate, and the averages were recorded. The limit of experimental errors on triplicates was systematically below 5%.

Note: full experimental conditions are systematically reported in the caption of the figures and tables.

The same batch process was used for the investigation of dye desorption and adsorbent recycling. The dye-loaded Tetra-Alg-CTAB adsorbents (recovered from sorption equilibrium tests) were first washed to remove the absorbed dye. In a second step, the adsorbent was mixed with 0.5 M HCl + 0.5 M NaCl mixed solution (adsorbent dose: 5.0 g adsorbent per 100 mL eluent) for 140 min. The suspension was centrifuged; the supernatant was analyzed for residual dye concentration. The collected sorbent was repeatedly rinsed with distilled water until the pH of the rinsing water reached 7.0. The regenerated sorbent was dried for 2 h at 40 °C in an oven (Gallenkamp BS Model OV-160, Loughborough (LE), UK) and subjected to 5 consecutive adsorption/desorption cycles. The adsorption (%) was calculated according to Equation (1).

## 4. Conclusions

Production of an adsorbent from algal biomass *Tetraselmis* sp. which was modified with a CTAB surfactant to produce Tetra-Alg-CTAB adsorbent as MO and MB adsorbents has been successfully carried out. The surface morphology of Tetra-Alg-CTAB illustrates that the grains attached to the surface produced by the CTAB surfactant modification efficiently enhanced adsorption capacity. Tetra-Alg-CTAB adsorbent was very effective in the adsorption of anionic MO dyes and less effective against cationic MB. MO adsorption by Tetra-Alg-CTAB was optimum at pH 10, a contact time of 90 min, and a concentration of 250 mg L^−1^. Tetra-Alg-CTAB adsorbent has a relatively large adsorption rate and capacity compared to the unmodified and Na^+^ ion-modified biomass (Tetra-Alg and Tetra-Alg-Na) towards the anionic MO dye. The best isotherm model was found to be the Langmuir isotherm, as it fits the adsorption experimental data well at equilibrium. On the other hand, the PSO model was found to be the most suitable adsorption kinetic model for this adsorption study. The reuse of Tetra-Alg-CTAB was carried out four times with adsorption percent > 70%, and the adsorbent was very effective in the adsorption of anionic dyes such as MO. These results proved that Tetra-Alg-CTAB could be employed as an alternative to commercial activated carbon for the removal of lead from the wastewater industry.

## Figures and Tables

**Figure 1 molecules-28-07839-f001:**
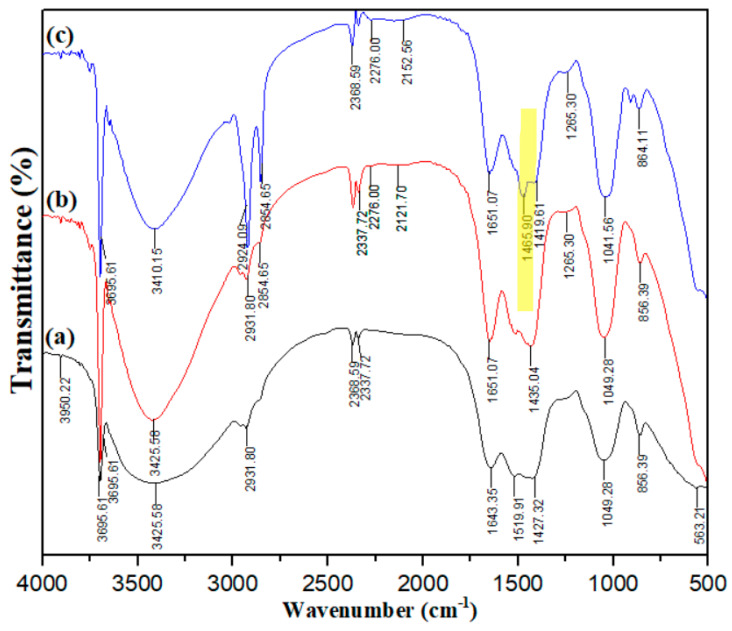
Spectra of IR from (**a**) Tetra-Alg, (**b**) Tetra-Alg-Na, and (**c**) Tetra-Alg-CTAB.

**Figure 2 molecules-28-07839-f002:**
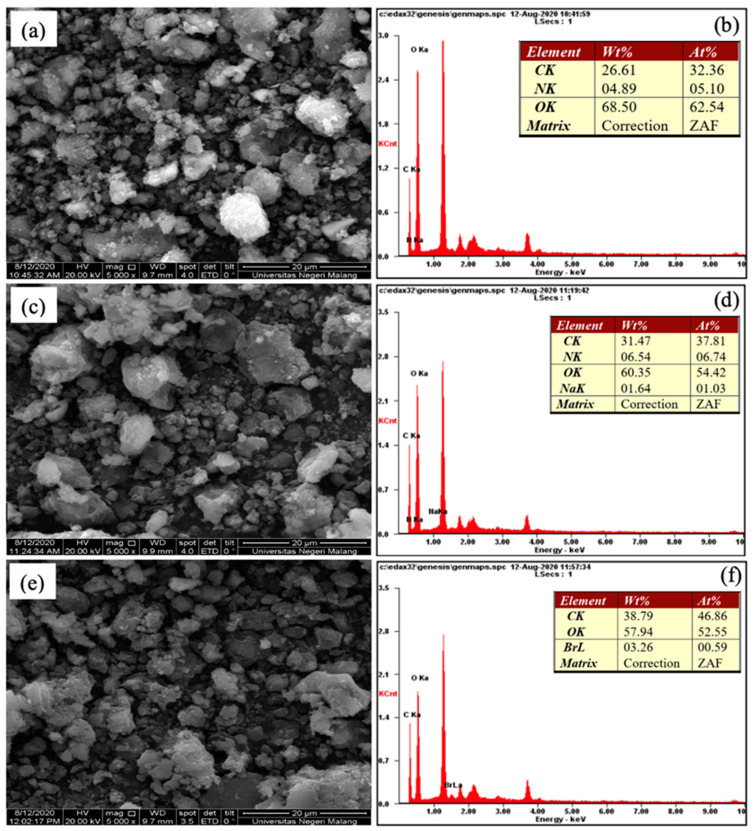
SEM images and EDX spectra of (**a**,**b**) Tetra-Alg, (**c**,**d**) Tetra-Alg-Na, and (**e**,**f**) Tetra-Alg-CTAB.

**Figure 3 molecules-28-07839-f003:**
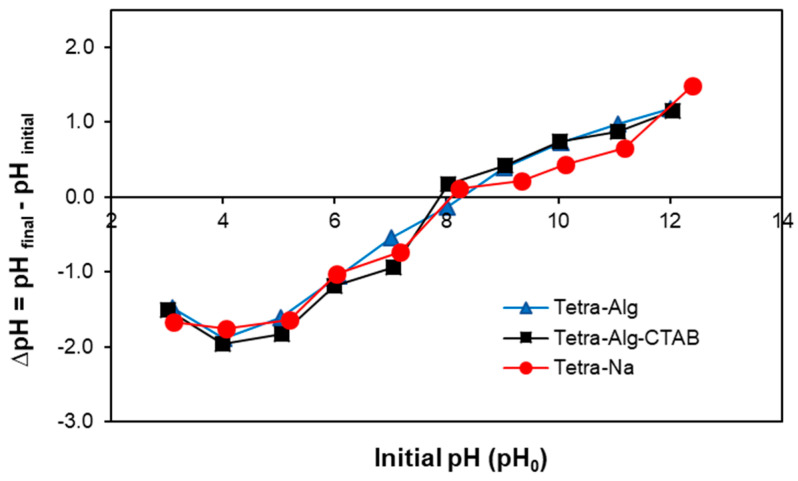
Determination of the point of zero charge of the Tetra-Alg, Tetra-Na, and Tetra-Alg-CTAB.

**Figure 4 molecules-28-07839-f004:**
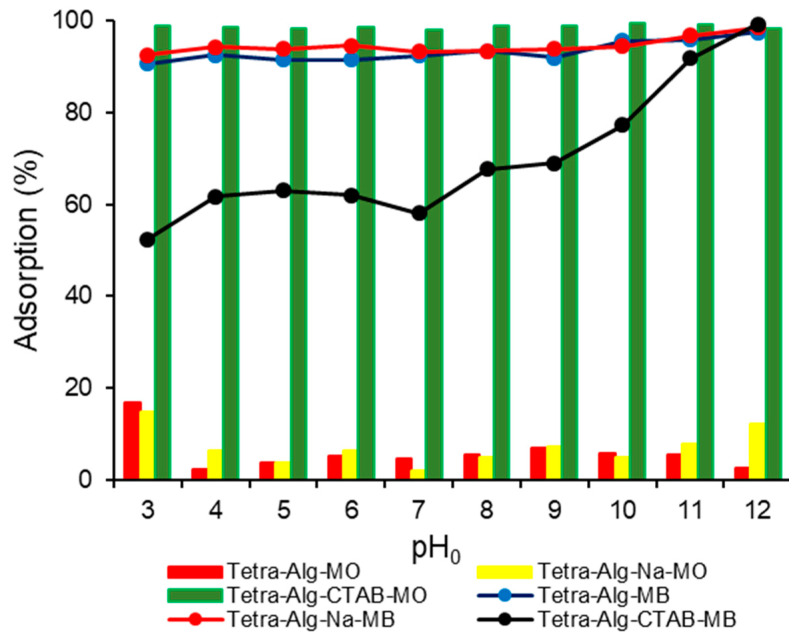
Effect of interaction pH_0_ on % adsorption of MO-dye and MB-dye by Tetra-Alg, Tetra-Alg-Na, Tetra-Alg-CTAB (adsorbent mass: 0.1 g; solution volume: 25 mL; C_0_: 10 mg L^−1^; temperature: 27 ± 1 °C; agitation speed: 150 rpm; time: 60 min).

**Figure 5 molecules-28-07839-f005:**
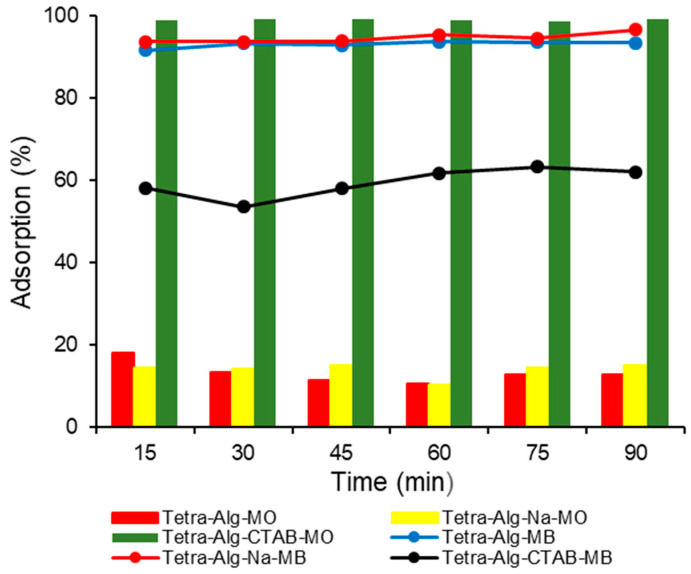
Effect of contact time on % adsorption of MO and MB by Tetra-Alg, Tetra-Alg-Na, Tetra-Alg-CTAB (adsorbent mass: 0.1 g; solution volume: 25 mL; C_0_: 10 mg L^−1^; temperature: 27 ± 1 °C; agitation speed: 150 rpm; pH: 10).

**Figure 6 molecules-28-07839-f006:**
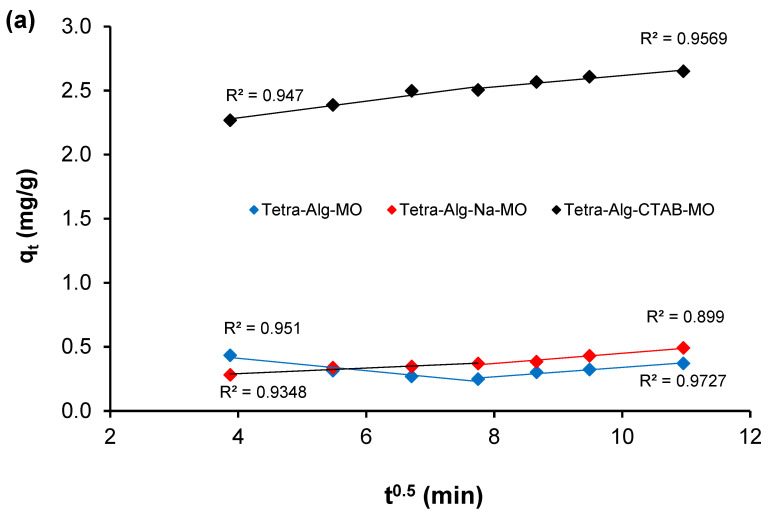
Intraparticle diffusion plot on adsorption of (**a**) MO and (**b**) MB on Tetra-Alg, Tetra-Alg-Na, Tetra-Alg-CTAB.

**Figure 7 molecules-28-07839-f007:**
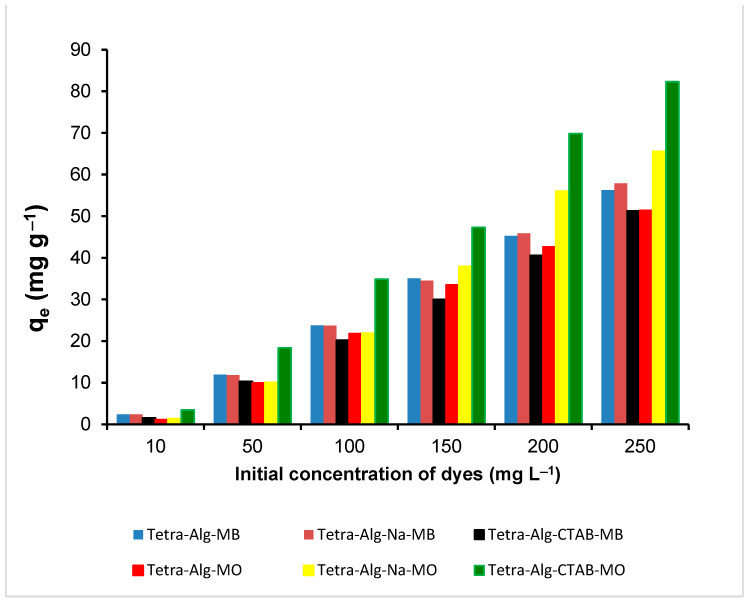
Effect of initial concentration on the amount of MO and MB adsorbed on Tetra-Alg, Tetra-Alg-Na, Tetra-Alg-CTAB (adsorbent mass: 0.1 g; solution volume: 25 mL; temperature: 27 ± 1 °C; agitation speed: 150 rpm; pH: 10; time: 90 min).

**Figure 8 molecules-28-07839-f008:**
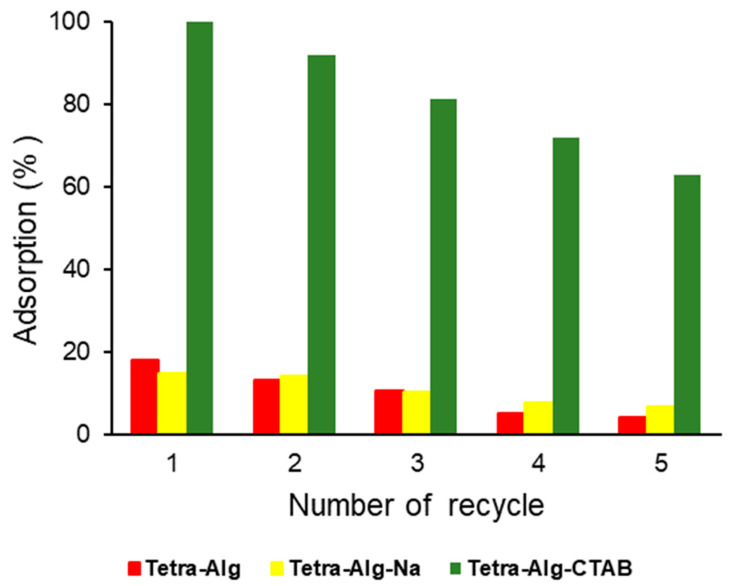
Reusability number efficiency of the adsorption of MO on Tetra-Alg, Tetra-Alg-Na, Tetra-Alg-CTAB.

**Table 1 molecules-28-07839-t001:** Pseudo-second-order kinetics for the adsorption of MO and MB on Tetra-Alg, Tetra-Alg-Na, Tetra-Alg-CTAB (adsorbent mass: 0.1 g; solution volume: 25 mL; *C*_0_: 10 mg L^−1^; temperature: 27 ± 1 °C; agitation speed: 150 rpm; pH: 10).

Adsorbent	Pseudo-Second-Order (PSO)							
	MO				MB			
	*q_e-exp_* (mg g^−1^)	*q_e-PSO_* (mg g^−1^)	*k*_2_ × 10^−3^ (g mg^−1^ min^−1^)	*R* ^2^	*q_e-exp_* (mg g^−1^)	*q_e-PSO_* (mg g^−1^)	*k*_2_ × 10^−3^ (g mg^−1^ min^−1^)	*R* ^2^
Tetra-Alg	0.301	0.290	6.032	0.999	2.307	2.340	0.758	0.999
Tetra-Alg-Na	0.357	0.357	9.551	0.994	2.407	2.363	0.749	0.999
Tetra-Alg-CTAB	2.479	2.475	12.640	0.999	1.467	1.445	0.473	0.997

**Table 2 molecules-28-07839-t002:** Intra-particle diffusion model for MO and MB adsorption on Tetra-Alg, Tetra-Alg-Na, Tetra-Alg-CTAB.

Adsorbent–Adsorbate	Initial Linear Portion			Second Linear Portion		
	*k_i_* _1_	*C* _1_	*R* _1_ ^2^	*k_i_* _2_	*C* _2_	*R* _2_ ^2^
	(mg g^−1^ min^0.5^)	(mg g^−1^)		(mg g^−1^ min^0.5^)	(mg g^−1^)	
Tetra-Alg-MO	0.005	0.603	0.951	0.037	0.029	0.899
Tetra-Alg-Na-MO	0.022	0.203	0.935	0.04	0.053	0.973
Tetra-Alg-CTAB-MO	0.065	2.028	0.947	0.044	2.172	0.957
Tetra-Alg-MB	0.059	2.024	0.951	0.030	2.261	0.900
Tetra-Alg-Na-MB	0.065	1.874	0.989	0.017	2.238	0.926
Tetra-Alg-CTAB-MB	0.037	1.164	0.947	0.024	1.287	0.940

**Table 3 molecules-28-07839-t003:** Adsorption isotherm parameters of MO and MB on Tetra-Alg, Tetra-Alg-Na, Tetra-Alg-CTAB (adsorbent mass: 0.1 g; solution volume: 25 mL; temperature: 27 ± 1 °C; agitation speed: 150 rpm; pH: 10; time: 90 min).

Adsorbent–Adsorbate	Adsorption Isotherm Parameters					
	Langmuir			Freundlich		
	*q_m_* (mg g^−1^)	*K_L_* × 10^−2^ (L mg^−1^)	*R* ^2^	*K_F_* × 10^−2^ ((mg g^−1^) (L mg^−1^)^1/*n*^)	*n*	*R* ^2^
Tetra-Alg-MO	51.631	0.361	0.946	14.894	0.310	0.985
Tetra-Alg-Na-MO	94.389	0.209	0.914	13.364	0.362	0.990
Tetra-Alg-CTAB-MO	128.369	3.520	0.933	18.059	0.235	0.994
Tetra-Alg-MB	50.055	3.395	0.915	0.407	1.187	0.995
Tetra-Alg-Na-MB	58.155	2.423	0.919	0.6.68	1.124	0.990
Tetra-Alg-CTAB-MB	51.013	0.239	0.933	0.224	0.707	0.991

**Table 4 molecules-28-07839-t004:** Comparison of adsorption capabilities of several adsorbents for the dyes MO and MB.

Adsorbent	Dyes	*q_m_* (mg g^−1^)	References
Cross-linked chitosan	MO	89.29	[48]
Chitosan/alumina composite	MO	33.00	[49]
Fe-La oxides co-loaded MgO (Fe-La/MgO) nanosheets	MO	30.38	[50]
Modified wheat straw 3.0	MO	50.40	[51]
Anchote peel	MO	103.03	[52]
Tetra-Alg-CTAB	MO	128.37	This research
Magnetite-loaded multi-wall carbon nanotubes	MB	48.00	[39]
Fe_3_O_4_@MIL-100(Fe) magnetite composite	MB	74.00	[44]
Silica-Polymer hybrid	MB	87.00	[53]
*Spirulina* sp. algae hybrid with a silica matrix	MB	74.00	[19]
Neolamarckia cadamba leaves	MB	101.40	[54]
Tetra-Alg-CTAB	MB	58.15	This research

## Data Availability

The data presented in this study are available upon request.

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
