# Peer review of "Cationic Surfactant-Modified Tetraselmis sp. for the Removal of Organic Dyes from Aqueous Solution"

_molecules, 2023, doi:10.3390/molecules28237839_

Round 1
Reviewer 1 Report
Comments and Suggestions for Authors
Removing nano-sized pollutants from water is an urgent problem in water treatment in various countries. Numerous scientific groups are working to solve this problem. In the current work, the authors use Tetraselmis sp. algae material for removing standard dyes methyl orange and methylene blue from water. The authors of the manuscript show that the dyeing of textiles causes a large amount of waste, which can be released into the environment with water. Among various methods of water purification, the authors highlight adsorption using algal biomass. To eliminate the disadvantages of algal biomass, it is proposed to use a cationic surfactant.
The authors begin to describe their results with structural data, using the methods of IR spectroscopy, SEM and EDX. Next, the authors identify the point of zero charge for all three samples. By changing the pH and type of Tetra-Alg adsorbent.
The best sorption activity was found for Tetra-Alg-CTAB at pH 10 for methyl orange with a concentration of 250 mg L-1. The optimal time was identified as 90 minutes. Regarding the reuse of Tetra-Alg-CTAB, the recommended number of cycles is 3 with an efficiency of more than 80%.
The conclusions of the manuscript include the main results and are presented at a decent level.
In general, the manuscript is easy to read and accessible. As a recommendation, a small editorial work could be carried out by a specialized specialist. Also, the work would become more valuable when comparing the results obtained with other sorption systems, possibly membranes.
L. 109. Typo in the table title - "dan"
L. 109. Typo in the table title - "dan"
Author Response
Reviewer 1
Removing nano-sized pollutants from water is an urgent problem in water treatment in various countries. Numerous scientific groups are working to solve this problem. In the current work, the authors use Tetraselmis sp. algae material for removing standard dyes methyl orange and methylene blue from water. The authors of the manuscript show that the dyeing of textiles causes a large amount of waste, which can be released into the environment with water. Among various methods of water purification, the authors highlight adsorption using algal biomass. To eliminate the disadvantages of algal biomass, it is proposed to use a cationic surfactant.
The authors begin to describe their results with structural data, using the methods of IR spectroscopy, SEM and EDX. Next, the authors identify the point of zero charge for all three samples. By changing the pH and type of Tetra-Alg adsorbent.
The best sorption activity was found for Tetra-Alg-CTAB at pH 10 for methyl orange with a concentration of 250 mg L-1. The optimal time was identified as 90 minutes. Regarding the reuse of Tetra-Alg-CTAB, the recommended number of cycles is 3 with an efficiency of more than 80%.
The conclusions of the manuscript include the main results and are presented at a decent level.
In general, the manuscript is easy to read and accessible. As a recommendation, a small editorial work could be carried out by a specialized specialist.
Re: Thank you very much for your appreciation of our work. We have improved the editing of our manuscript according to your helpful comments. We hope our revised manuscript meets your requirements.
Also, the work would become more valuable when comparing the results obtained with other sorption systems, possibly membranes.
Re: the comparison of the investigated biosorbent with other sorbents is investigated in Table 5.
The following text has been added to page 13, lines 397-404.
“The comparison of sorption performances for Tetra-Alg-CTAB and with other algal clearly demonstrates the synergistic effect of precursors. The incorporation of the cationic surfactant CTAB in the algal biomass allows for the improvement of the sorption of MO. The cationic surfactant-modified biosorbent exhibited a notable increase in its ability to attract anionic dye molecules. This is due to the introduction of positive charges on the Tetra-Alg-CTAB surface, which enhances the adsorption of MO via electrostatic interaction, while MB sorption is more likely to occur through cation exchange.”
109. Typo in the table title - "dan"
Re: Done, it has been corrected, then Table one was moved to the supplementary material as suggested by the second reviewer, thank you.
Reviewer 2 Report
Comments and Suggestions for Authors
1. The modification method is effective for the enhancement of adsorption capacity of biomass, but the related mechanism was not described in the abstract.
2. In my opinion, the prepared materials showed interesting adsorption performance. Such performance was tested by using dyes, and optimized with different conditions. Thereby, in the introduction, many sentences focused on the dyes-bearing wastewater and the related pollution, and should be concisely described. Instead, the upcycling of biomass, especially, Tetraselmis sp. Algae, should be emphasized.
3. Section 2.1, the table 1 can be removed. Such information can be described in words. The equations 1-3 should be removed.
4. Figure 1, the new peaks derived from the modification, should be highlighted. Some data in Figure 3 was not significant, and thereby the corresponding error bar should be added.
5. Given that the MB and MO were used for the determination of biomass capacity, the related calculation should be taken. For instance, the surface charge and density, the surface area, can be calculated by using the adsorption data, especially for MB.
6. the pH value is very important for the adsorption, and the related mechanism should be discussed.
7. Actually, the prepared biomass was often used for the selective adsorption of contaminants from wastewater, but with the use, the adsorption capacity steadily decreased. What is the major reason?
Comments on the Quality of English Languagereadable
Author Response
Reviewer 2
- The modification method is effective for the enhancement of adsorption capacity of biomass, but the related mechanism was not described in the abstract.
Re: Done, the following text has been added to the abstract.
“Due to the modification with the cationic surfactant CTAB, anionic dyes can be strongly sorbed in alkaline pH due to strong electrostatic attraction, while MB is more likely to involve cation exchange and hydrogen bonding.”
- In my opinion, the prepared materials showed interesting adsorption performance. Such performance was tested by using dyes, and optimized with different conditions. Thereby, in the introduction, many sentences focused on the dyes-bearing wastewater and the related pollution, and should be concisely described. Instead, the upcycling of biomass, especially, Tetraselmis sp. Algae, should be emphasized.
Re: Done the following text has been added to the introduction section to refer to the upcycling of the exhausted biomass, please see page 3, lines 92-100 of the revised manuscript.
“A good sorbent, especially for large-scale applications, must be recyclable in order to be competitive. Both the demand for fresh sorbent and the issue of disposing of spent sorbent can be addressed via regeneration. Different regeneration techniques have been applied, with varying degrees of effectiveness. Solvent washing, thermal, chemical, and electrochemical regeneration are some of these techniques [1]. Recently upcycling of dyes-laden spent sorbent via an appropriate management pathway (anaerobic fermentation) for the development of a low-cost hydrogen production scenario was investigated [2], the findings of that study inferred that the spent algal sorbent could be introduced as an effective substitute for biohydrogen production.”
Also see page 3, lines 105-106.
“Dye desorption and sorbent recycling are carried out after sorption kinetic and equilibrium investigations”
Section 2.1, the table 1 can be removed. Such information can be described in words. The equations 1-3 should be removed.
Re: Table 1 has been moved to supplementary material; however, we prefer to keep equations 1-3 to give the readers a better understanding of the calculated adsorption parameters used in this study.
- Figure 1, the new peaks derived from the modification, should be highlighted. Some data in Figure 3 was not significant, and thereby the corresponding error bar should be added.
Re: Done, the new peak derived from the modification with CTAB is highlighted with yellow color. The data values in Figure 3 as well as Figures 4, 5, 7, and 8 were taken from triplicate analysis, the reported values were taken from the average of three measurements, The limit of experimental errors on triplicates was systematically below 5%. The following text has been added to the manuscript. Please see page 4, lines 158- 160 of the revised manuscript. Also, full experimental conditions are systematically reported in the caption of the figures and tables.
“All adsorption tests were performed in triplicate, and the averages were recorded. The limit of experimental errors on triplicates was systematically below 5%.”
- Given that the MB and MO were used for the determination of biomass capacity, the related calculation should be taken. For instance, the surface charge and density, the surface area, can be calculated by using the adsorption data, especially for MB.
Re: The adsorbent surface charge was calculated according to the pH change draft method, however, the surface area of the adsorbent was calculated using MB adsorption data. The following text has been added to the revised manuscript, please see page 6-7, lines 233-240.
“The surface area of the resin was calculated using the following equation [3]:
(4)
whereas is the adsorbent surface area in m2 g-1, G the amount of methylene blue adsorbed (g) based on Langmuir adsorption capacity, NAv the Avogadro's number (6.02 × 1023), Ø the methylene blue molecular cross-section (197.2 Å2), MW the molecular weight of methylene blue (373.9 g mol-1) and M is the mass of adsorbent (g). The surface area of Tetra-Alg, Tetra-Alg-Na, and Tetra-Alg-CTAB was found to be 158.93, 184.64, and 161.97 m2 g-1, respectively.”
- The pH value is very important for the adsorption, and the related mechanism should be discussed.
Re: Yes, the pH has a significant impact on the adsorption process, as it alters both adsorbent active surface charge and dissolved adsorbate speciation, The following text has been added to the revised manuscript, please see page 8, lines 268-276.
“Therefore, the increase in MO removal after cationic surfactant modification can be explained by electrostatic interactions. Even at pH0 values = 3, the adsorption of MO remained significantly high due to the presence of the cationic surfactant. while MB adsorption is more likely to involve cation exchange and hydrogen bonding. The adsorption process mechanism may also be significantly influenced by other forces such as n–π stacking interactions, and dipole-dipole hydrogen bonding [4]. It's also important to remember that some oppositely charged surface sites may have existed concurrently with the overall negatively or positively charged adsorbent surface charge.”
- Actually, the prepared biomass was often used for the selective adsorption of contaminants from wastewater, but with the use, the adsorption capacity steadily decreased. What is the major reason?
Re: Regeneration studies are essential to evaluate the reusability adsorbents for practical applications due to stringent ecological and economic demand for sustainability, usually desorption process occurs in an acidic medium which results in the destruction of the sorbent biomass with succsive adsorption/desorption cycles.
Round 2
Reviewer 2 Report
Comments and Suggestions for Authors
no comments.
Comments on the Quality of English Languageno comments.